# OpenReview forum: "Scalable Fingerprinting of Large Language Models"
_ICLR.cc/2025/Workshop/BuildingTrust — BuildingTrust_

### Official Review · Reviewer_diBH · 2025-02-20

**Rating:** 5
**Confidence:** 3

**Review:**

This paper proposes a novel scheme for generating and inserting fingerprints in LLMs. Employing the Perinucleus sampling method for fingerprint generation and leveraging model averaging and data - mixing regularization techniques for training, it notably boosts the number of insertable fingerprints. The scheme has been verified to ensure key fingerprint properties like harmlessness, persistence, and scalability.

**Questions:**

1. The experiments are solely based on the Llama-series Base models (1B, 3B, 8B). How effective is the method on models with diverse architectures and scales?
2. The information about the SFT experimental settings is not clearly described. Is training for only 2 epochs on the alpaca dataset sufficient?
3. How effective is the proposed method when fully fine - tuned in a downstream specific domain?
4. The paper claims that more fingerprints are better. However, how does the model's computational complexity change during inference as the number of inserted fingerprints increases?
5. What impact will the proposed method face when the fingerprinted model undergoes compression or quantization operations?

---

### Official Review · Reviewer_AYJG · 2025-03-01
**Review of Scalable Fingerprinting of Large Language Models**

**Rating:** 6
**Confidence:** 4

**Review:**

### **Summary**
This paper focuses on the **scalability** of model fingerprinting, introducing **Perinucleus Sampling**, a novel method that allows embedding a significantly higher number of fingerprints compared to prior approaches without degrading model performance. The study also highlights security concerns such as collusion attacks, discussing how scalability can help mitigate these threats.

### **Strengths**
The primary contribution of this work is its ability to insert a **substantially larger number of fingerprints** without a major drop in model utility, addressing a critical limitation of prior fingerprinting schemes. The authors conduct **sufficient comparisons** with existing methods to illustrate that increasing the number of fingerprints does not significantly degrade performance. Additionally, they show that their approach **achieves higher persistence** when the model is fine-tuned on a different dataset, which is a crucial property for real-world deployment.

### **Weaknesses**
The **main limitation** of the work is the **lack of comprehensive empirical evaluations on the security aspects** of scalability. While the **theoretical discussion is strong**, there is **insufficient empirical validation**. Moreover, Appendix E lacks direct comparisons with baseline methods under adversarial conditions, making it unclear how the proposed method fares against existing methods under similar adversarial settings. Incorporating more empirical results under different attack scenarios and comparing against other fingerprinting techniques in these settings would greatly strengthen the security claims of the paper.

Overall, while the paper makes a notable contribution to scalable fingerprinting with strong theoretical grounding and empirical validation on persistence and harmlessness.

---

### Official Review · Reviewer_LWPD · 2025-03-02
**Interesting work on LLM fingerprint scalability**

**Rating:** 8
**Confidence:** 5

**Review:**

This paper focuses on improving the scalability of model fingerprinting for large language models (LLMs). The paper proposes a new method, Perinucleus sampling, which aims to embed a large number of fingerprints into LLMs without significantly degrading their utility.

Pros:
- The paper argues that scalability of fingerprints is critical, because proving ownership via fingerprints involves revealing the fingerprint used. This is a novel perspective that has been rarely discussed. In Section 5 the paper exemplifies how scalability is necessary for security with detailed analyses.
- The paper demonstrates that even with a large number of fingerprints, the degradation in model performance is minimal. This balance between uniqueness and harmlessness is crucial for practical deployment.

Cons:
- Since the method uses a large number of fingerprints, how does the fingerprint training time scale? Does it require significantly more training time compared to other methods? The paper has not addressed this issue.

---

### Decision · Program_Chairs · 2025-03-04

Accept